# Synergistic Inhibitory Effect of Polymyxin B in Combination with Ceftazidime against Robust Biofilm Formed by *Acinetobacter baumannii* with Genetic Deficiency in AbaI/AbaR Quorum Sensing

Yinyue Li,[a] Bo Wang,[b] Feng Lu,[a,d] Juhee Ahn,[c] Wenwen Zhang,[a] Liangliang Cai,[a] Jiahui Xu,[a] Yi Yin,[a] Qingchao Cao,[a] Zhenyu Ren,[a] Xinlong He[a,d,e,f]

[a]Department of Pathogen Biology, School of Medicine, Yangzhou University, Yangzhou, People's Republic of China
[b]Department of Clinical Laboratory, First Affiliated Hospital of Anhui Medical University, Hefei, Anhui, People's Republic of China
[c]Department of Medical Biomaterials Engineering, College of Biomedical Science, Kangwon National University, Chuncheon, Gangwon, South Korea
[d]Jiangsu Key Laboratory of Experimental and Translational Non-coding RNA Research, Yangzhou University, Yangzhou, People's Republic of China
[e]Jiangsu Key Laboratory of Zoonosis, Yangzhou University, Yangzhou, People's Republic of China
[f]Jiangsu Co-Innovation Center for the Prevention and Control of Important Animal Infectious Diseases and Zoonosis, College of Veterinary Medicine, Yangzhou University, Yangzhou, People's Republic of China

Yinyue Li and Bo Wang contributed equally to this article. Author order was determined randomly.

**ABSTRACT** Carbapenem resistance of *Acinetobacter baumannii* poses challenges to public health. Biofilm contributes to the persistence of *A. baumannii* cells. This study was designed to investigate the genetic relationships among carbapenem resistance, polymyxin resistance, multidrug resistance, biofilm formation, and surface-associated motility and evaluate the antibiofilm effect of polymyxin in combination with other antibiotics. A total of 103 clinical *A. baumannii* strains were used to determine antibiotic susceptibility, biofilm formation capacity, and motility. Enterobacterial repetitive intergenic consensus (ERIC)-PCR fingerprinting was used to determine the genetic variation among strains. The distribution of 17 genes related to the resistance-nodulation-cell division (RND)-type efflux, autoinducer-receptor (AbaI/AbaR) quorum sensing, oxacillinases (OXA)-23, and insertion sequence of IS*Aba1* element was investigated. The representative strains were chosen to evaluate the gene transcription and the antibiofilm activity by polymyxin B (PB) in combination with merapenem, levofloxacin, and ceftazidime, respectively. ERIC-PCR-dependent fingerprints were found to be associated with carbapenem resistance and multidrug resistance. The presence of *bla*~OXA-23~ was found to correlate with genes involved in IS*Aba1* insertion, AbaI/AbaR quorum sensing, and AdeABC efflux. Carbapenem resistance was observed to be negatively correlated with biofilm formation and positively correlated with motility. PB in combination with ceftazidime displayed a synergistic antibiofilm effect against robust biofilm formed by an *A. baumannii* strain with deficiency in AbaI/AbaR quorum sensing. Our results not only clarify the genetic correlation among carbapenem resistance, biofilm formation, and pathogenicity in a certain level but also provide a theoretical basis for clinical applications of polymyxin-based combination of antibiotics in antibiofilm therapy.

**IMPORTANCE** Deeper explorations of molecular correlation among antibiotic resistance, biofilm formation, and pathogenicity could provide novel insights that would facilitate the development of therapeutics and prevention against *A. baumannii* biofilm-related infections. The major finding that polymyxin B in combination with ceftazidime displayed a synergistic antibiofilm effect against robust biofilm formed by an *A. baumannii* strain with genetic deficiency in AbaI/AbaR quorum sensing further

Address correspondence to Xinlong He, hexldragon@hotmail.com.

The authors declare no conflict of interest.

provides a theoretical basis for clinical applications of antibiotics in combination with quorum quenching in antibiofilm therapy.

**KEYWORDS** *Acinetobacter baumannii*, carbapenem resistance, polymyxin B, antibiofilm formation, quorum sensing

A*cinetobacter baumannii* is an important opportunistic pathogen causing various nosocomial infections, including skin and soft tissue infections (1), urinary tract infections (2), pneumonia (3), and even bacteremia (4). With the extensive use and especially the abuse of antibiotics, resistant bacteria emerged quickly, and multidrug-resistant (MDR) (5) *A. baumannii* has become widespread. Carbapenem antibiotics with broad antibacterial spectrum and high bactericidal activity are the first line of treatment for severe infections caused by MDR *A. baumannii*. However, carbapenem-resistant *A. baumannii* (CRAB) has become one of the most common nosocomial pathogens, especially in the intensive care unit (ICU) (6–8). The frequent emergence of CRAB, especially those with high virulence, poses severe challenges to public health (9, 10). The interplay between genetic virulence regulation and carbapenem resistance seems to be strain dependent (11). Molecular typing plays a key role in understanding the basic mechanism of *A. baumannii* infection and discovering the relationship between bacterial species (12).

*A. baumannii* resistance to carbapenems is more closely related to the production of oxacillinase (OXA)-type carbapenemases (13). Among these D class $\beta$-lactamases, OXA-23 is the most common carbapenemase produced by CRAB in recent years (14). The insertion element IS*Aba1* is essential for high-level production of OXA-23 (15). The widely distributed resistance-nodulation-cell division (RND) family efflux pumps, including AdeABC, AdeIJK, and AdeFGH, are also important factors that have to be considered for carbapenem resistance in *A. baumannii* (16). In addition, the biofilm-forming capacity of *A. baumannii* has been thought to play an essential role in the persistence of *A. baumannii* cells under severe environmental conditions (17). Surface-associated motility of *A. baumannii* is likely to be mediated by type IV pili, an important virulence factor involved in bacterial pathogenicity (18). Similar to the typical LuxI/LuxR (19), AbaI/AbaR quorum sensing plays an important role in maintaining the expression of $bla_{OXA-51}$ and *ampC* resistance genes, as well as the carbapenem resistance phenotypes in *A. baumannii* (20). The AbaI/AbaR system is also required for the regulation of biofilm formation and motility (21). However, the genetic relationships among carbapenem resistance, MDR, quorum sensing, biofilm formation, and motility remain unclear.

At the time when carbapenem resistance was widespread, polymyxins were particularly important as the last line of defense in the treatment of multidrug-resistant bacterial infections. Low-dose administration of meropenem, levofloxacin, and tigecycline displays increased risk of biofilm-associated infections (22, 23). Although polymyxin in combination with these conventional antibiotics has partially shown a pleasing synergistic inhibitory effect on the proliferation of *A. baumannii* cells (24), its effect on biofilm formation is still unclear. This study was designed to investigate the genetic correlation among carbapenem resistance, polymyxin resistance, MDR, AbaI/AbaR quorum sensing, motility, and biofilm formation and evaluate the potential antibiofilm effect of polymyxin in combination with meropenem, levofloxacin, and ceftazidime against *A. baumannii*.

## RESULTS AND DISCUSSION

**ERIC-PCR-dependent fingerprints of *A. baumannii* are associated with carbapenem resistance and multidrug resistance.** According to Clinical and Laboratory Standards Institute (CLSI), the minimum inhibitory concentration (MIC) is visually defined as the absence of growth in cation-adjusted Mueller-Hinton (MH) medium. In this study, we found that there was no significant difference between the MIC values obtained by using MH and nutrient broth (NB) media and for antibiotics of meropenem (MEM), levofloxacin (LEV),

ceftazidime (CAZ), and polymyxin B (PB), respectively. Enterobacterial repetitive intergenic consensus (ERIC)-PCR fingerprinting has been demonstrated to be a reliable technique for discriminating intraspecific variations (25). Using ERIC-PCR analysis, a total of 103 *A. baumannii* strains were classified into nine ERIC-PCR-dependent genotypes showing four groups (a group contains more than three strains) with types of II ($n$ = 53, 51.5%), III ($n$ = 20, 19.4%), IV ($n$ = 11, 10.7%), and V ($n$ = 13, 12.6%); two strains with type of I; and four single strains with type of VI, VII, VIII, and IX, respectively (Fig. S1). The bacterial sources mostly came from sputum ($n$ = 70, 68%) followed by wounds ($n$ = 19, 18.5%) (Fig. S2). Among sputum and wound-derived strains, carbapenem-resistant strains accounted for 72.9 and 78.9%, respectively, which remained at the same level as the 75.1% resistance rate of the previous year (26). This observation suggests that respiratory tract infections and wound infections caused by CRAB are still burdens for health care in China. The type II and III strains were observed to be dominated by carbapenem-resistant strains (Fig. 1), showing 96.2 and 95% resistance rates (Table 1), respectively, while the IV and V type strains were mainly carbapenem-sensitive strains (72.7 and 84.6%). These observations suggest that the carbapenem resistance of *A. baumannii* is intrinsically related to specific genotypes. The ERIC-dependent fingerprints can be used as a useful genetic marker for the epidemiological investigation of carbapenem-resistant *A. baumannii*.

As shown in Fig. 1, most of the type II and III strains also displayed resistance to LEV and CAZ, and more than 70% of type II and III strains showed MDR to MEM, LEV, and CAZ (Table 1). PB is used as the last-line drug for the treatment of multidrug-resistant bacterial infections. A recent meta-analysis has shown that the overall frequency of polymyxin-resistant *A. baumannii* in hospitals worldwide is 13%, and the average incidence of infections by polymyxin-resistant *A. baumannii* in hospitalized patients from 2012 to 2017 in Eastern Asian countries, including China, is 18% (27). As shown in Table 1, there was no genotype difference in the distribution of PB-resistant strains. In mechanism, resistance to carbapenems, quinolones, and cephalosporins in *A. baumannii* is primarily mediated by mobile genetic elements (28). Plasmids that carry resistance genes present in this bacterium usually carry genes that encode low susceptibility to more than one class of antibiotics (29). Resistance to polymyxins in *A. baumannii* is basically mediated by chromosomal mutations (30), which hardly correlate with resistance to other antibiotics. Despite the advancement of *mcr* as a gene encoding plasmid-mediated resistance to polymyxins in nonfermenting Gram-negatives such as *A. baumannii*, it still represents a limited mechanism in this group of pathogens compared to chromosomal mutations in the *pmrCAB* and *lpxACD* operons (30). It should be noted that the PB resistance rate (41.8% in average) was much higher than the world average in recent years. In addition, the multidrug-resistant CRAB that was resistant to PB appeared in different genotypes, and its proportion in type II strains was even as high as 26% (Table 1). Therefore, the development of antibiotic resistance of *A. baumannii* in China is not a positive sign. From the perspective of effective treatment of MDR *A. baumannii* infections and curbing the development of antibiotic resistance, our observations stress the urgency of new drug development and the importance of current polymyxin combination therapy.

**The presence of *bla*$_{OXA-23}$ is genetically correlated with IS*Aba1* insertion, AbaI/AbaR quorum sensing, and AdeABC efflux.** In this study, 85.3% of MEM-resistant and 100% of MEM-intermediate *A. baumannii* strains were observed to harbor *bla*$_{OXA-23}$ gene (Fig. 1). The *bla*$_{OXA-23}$ gene was detected in more than 85% of type II and III strains (Table 2), which is consistent with the carbapenem-resistance phenotype. All *bla*$_{OXA-23}$-harboring *A. baumannii* strains were also observed to contain IS*Aba1* element (Fig. 1). Data analysis also showed a positive correlation in gene presence between *bla*$_{OXA-23}$ and IS*Aba1* (Table 3), suggesting a genetic correlation between *bla*$_{OXA-23}$ and IS*Aba1*. The IS*Aba1* element is commonly upstream of the *bla*$_{OXA-23}$ (31), which forms a neighbor relation in genome structure that is crucial for both the horizontal transfer of *bla*$_{OXA-23}$ (32) and the vertical inheritance of IS*Aba1*. Quorum sensing has been proven to be important for success of an intraspecific conjugation of *Agrobacterium* (33) and an interspecific conjugation between *Escherichia coli* and *Pseudomonas aeruginosa*

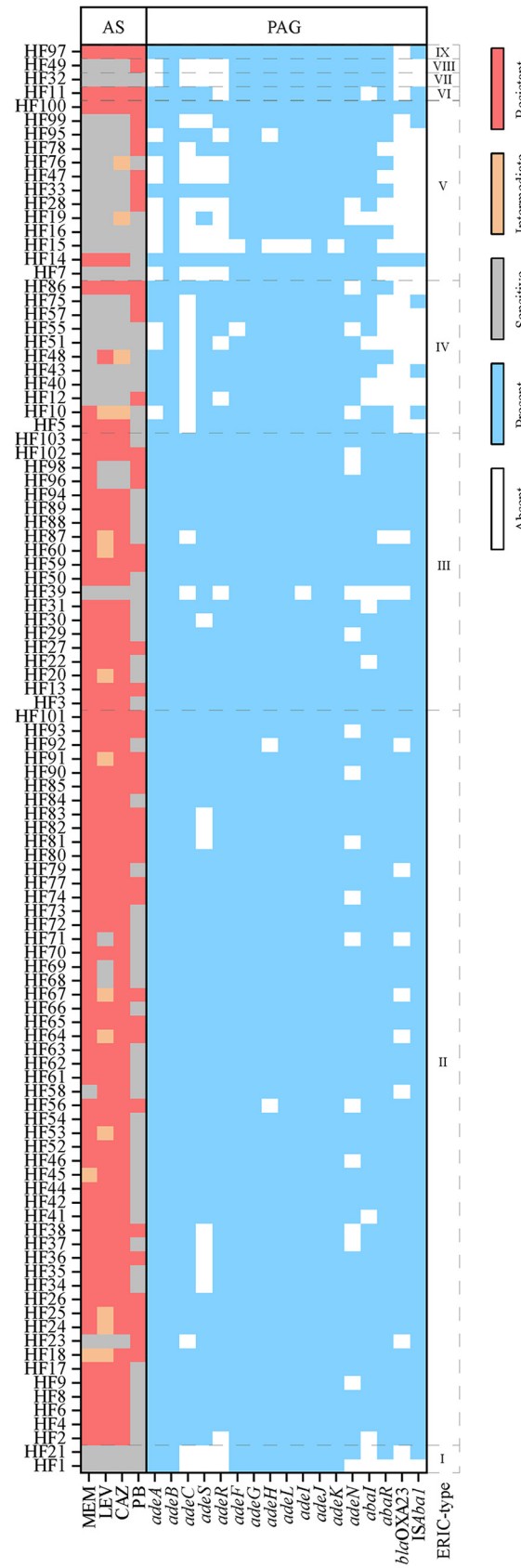

**FIG 1** Heat map of antibiotic resistance and gene presence in *Acinetobacter baumannii* strains. The roman numerals on the right (I to IX) indicate the different clades identified from the ERIC-PCR-

(34). On account of a correlated presence of $bla_{OXA-23}$ and *abaR* (Table 3), AbaI/AbaR quorum sensing might also be associated with the horizontal transfer of $bla_{OXA-23}$. OXA genes are widespread in clinic *A. baumanii*; most of them are $bla_{OXA-23}$ and $bla_{OXA-51}$ (35). IS*Aba1* insertion could contribute to high-level expression of $bla_{OXA-51}$, as well as $bla_{OXA-23}$ (36). The lack of consideration of $bla_{OXA-51}$ is a limitation of this study.

Overexpression of RND family efflux systems is generally considered to play a major role in the MDR of *A. baumannii* (37). Except for *adeB* and *adeG*, all other genes of RND-type efflux systems were found to have different degrees of deficiency in type II to V strains (Fig. 1), and the missing rate of *adeC* was found to be even higher than 90% in IV type strains (Table 2). Strains with a complete efflux system accounted for more than 80% of type II and type III strains, and the distribution of AdeFGH in type III strains even reached 100% (Table 2). The completeness of these efflux systems in IV and V strains was relatively low, and the AdeABC efflux system was the most prominent, showing proportions of 9.1 and 23.1% in IV and V strains (Table 2). The significant difference in the distribution of the AdeABC system in the type II, III, IV, and V strains is likely to be the main reason for the significant difference in the distribution of multidrug resistance in different genotype strains in this study. The $bla_{OXA-23}$ was found to coexist with genes related to AdeABC efflux system, showing correlation coefficient as high as 0.796 ($P < 0.01$) (Table 3). The expression of $bla_{OXA-23}$ was also observed to correlate with both *adeB* and *adeJ* at an mRNA level (Fig. 2B). These observations provide a reasonable explanation for the phenomenon that CRAB strains commonly exhibit the MDR phenotype (38). IS*Aba1* element is thought to contribute to the carbapenem resistance by affecting not only the expression of $\beta$-lactamases but also the regulation of RND-type efflux (15, 39). The expression of *adeABC* has been reported to be regulated by a two-component signaling system encoded by *adeR* and *adeS* (40), while the expression of *adeFGH* and *adeIJK* is regulated by *adeL* (41) and *adeN* (42), respectively. Mutation in *adeS* by IS*Aba1* insertion is associated with an increased expression of AdeABC, as well as a decreased susceptibility to tigecycline in *Acinetobacter* spp. (39, 43). An insertion of IS*Aba1* in *adeN* has been previously reported (44), which might have led to diminished susceptibility to antibiotics that are substrates for the AdeIJK. In this study, structurally, 40.3% of MDR *A. baumannii* strains were found to harbor *adeN* with an insertion of IS*Aba1* ($adeN^{\wedge IS Aba1}$) at a locus between 298 and 299 (Fig. S3). A correlated presence of IS*Aba1* with *adeS*, *adeR*, and *adeL* (Table 2) might increase the risk of overexpression of these RND-type efflux systems, resulting in MDR.

**Carbapenem resistance is negatively correlated with biofilm formation and positively correlated with motility.** The ability to form biofilms is closely related to the persistence of bacteria in unfavorable environments and is considered the main cause of chronic infections (45). Motility has been linked to increased pathogenicity in various bacteria, including *A. baumannii* (46, 47). Reduced fitness and virulence have been demonstrated to be associated with increased resistance to certain antibiotics, including polymyxins (11, 48) and ciprofloxacin (49). There were no significant differences in biofilm and motility among strains of different genotypes (II to V) (data not shown). However, in general, as shown in Table 3, both the carbapenem-resistant phenotype and the presence of $bla_{OXA23}$ were significantly negatively correlated with the formation of biofilms ($P < 0.01$), and the presence of $bla_{OXA-23}$ that was negatively correlated with the biofilm formation was positively correlated with motility ($P < 0.05$). An inverse correlation between motility and biofilm formation provides important information for the choice of antimicrobial and antibiofilm therapy of infections. In this study, the presence of IS*Aba1* that was negatively correlated with the biofilm formation was also found to be positively correlated with the motility (Table 3). Mutation in *fpvA* (a TonB re-

**FIG 1** Legend (Continued)
dependent fingerprints. Breakpoints were defined for ceftazidime as $\leq 8$, 16, and $\geq 32$ $\mu$g/mL; for levofloxacin as $\leq 2$, 4, and $\geq 8$ $\mu$g/mL; for meropenem as $\leq 4$, 8, and $\leq 16$ $\mu$g/mL; and for polymyxin B as $\leq 2$ and $\geq 4$ $\mu$g/mL for designating strains as antibiotic sensitive, intermediate (if available), and resistant, respectively. AS, antibiotic susceptibility; PAG, presence or absence of gene detected; MEM, meropenem; LEV, levofloxacin; CAZ, ceftazidime; PB, polymyxin B.

**TABLE 1** Proportion of antibiotic-resistant *A. baumannii* strains[a]

| Antibiotic | Proportion of antibiotic-resistant strains (%) | | | |
| | ERIC type II | ERIC type III | ERIC type IV | ERIC type V |
|---|---|---|---|---|
| MEM | 92.5 | 95 | 27.3 | 15.4 |
| LEV | 79.2 | 70 | 27.3 | 15.4 |
| CAZ | 98.1 | 85 | 18.2 | 15.4 |
| PB | 41.5 | 35 | 36.4 | 53.8 |
| Multidrug | | | | |
| MEM, LEV, CAZ | 75.5 | 70 | 18.2 | 15.4 |
| PB, MEM, LEV, CAZ | 26.4 | 20 | 9.1 | 7.7 |

[a]MEM, meropenem; LEV, levofloxacin; CAZ, ceftazidime; PB, polymyxin B.

ceptor homologue) has been demonstrated to be associated with an accelerated motility and a decreased production of extracellular polysaccharides and quorum-sensing molecules in *Pseudomonas syringae* (50). Thus, a disruption of TonB-dependent siderophore receptor gene that can be caused by IS*Aba1* insertion (44) could be involved in an upregulation of motility and a concurrent downregulation of biofilm formation observed in this study. This study also establishes a link between IS*Aba1* and AbaI/AbaR quorum sensing on account of a correlated presence of IS*Aba1* and *abaR*. However, the inner connection between IS*Aba1* and AbaI/AbaR needs to be further clarified.

**Synergistic antimicrobial effects of PB in combination with antibiotics against *A. baumannii*.** Since the return of polymyxins as effective drugs for the treatment of multidrug-resistant bacterial infections, especially carbapenem-resistant Gram-negative bacterial infections, the emergence of PB-resistant *A. baumannii* has increased in recent years (51, 52). In this study, the antimicrobial effect of PB in combination with

**TABLE 2** Proportion of *A. baumannii* strains with the presence of genes

| Gene | Proportion of *A. baumannii* strains with genes (%) | | | |
| | ERIC type II | ERIC type III | ERIC type IV | ERIC type V |
|---|---|---|---|---|
| Carbapenemases | | | | |
| *bla*~OXA23~ | 86.8 | 90 | 0 | 15.4 |
| Insertion sequence | | | | |
| IS*Aba1* | 100 | 100 | 27.3 | 23.1 |
| AdeABC system | | | | |
| *adeA* | 100 | 100 | 72.7 | 38.5 |
| *adeB* | 100 | 100 | 100 | 100 |
| *adeC* | 98.1 | 90 | 9.1 | 30.8 |
| *adeS* | 84.9 | 95 | 100 | 46.2 |
| *adeR* | 98.1 | 95 | 81.8 | 38.5 |
| *adeA, adeB, adeC, adeS, adeR* | 83 | 85 | 9.1 | 23.1 |
| AdeFGH system | | | | |
| *adeF* | 100 | 100 | 90.9 | 92.3 |
| *adeG* | 100 | 100 | 100 | 100 |
| *adeH* | 96.2 | 100 | 100 | 84.6 |
| *adeL* | 100 | 100 | 100 | 92.3 |
| *adeF, adeG, adeH, adeL* | 96.2 | 100 | 90.9 | 84.6 |
| AdeIJK system | | | | |
| *adeI* | 100 | 95 | 100 | 92.3 |
| *adeJ* | 100 | 95 | 100 | 100 |
| *adeK* | 100 | 95 | 100 | 92.3 |
| *adeN* | 81.1 | 80 | 72.7 | 84.6 |
| *adeI, adeJ, adeK, adeN* | 81.1 | 80 | 72.7 | 76.9 |
| Quorum sensing | | | | |
| *abaI* | 96.2 | 85 | 72.7 | 92.3 |
| *abaR* | 100 | 90 | 36.4 | 53.8 |
| *abaI, abaR* | 96.2 | 80 | 36.4 | 53.8 |

**TABLE 3** Correlation matrix of Spearman correlation coefficients between gene presence, antibiotic resistance, biofilm formation, and motility in *A. baumannii*

| Factor | bla$_{OXA-23}$ | ISAba1 | adeA/B/C | adeS | adeR | adeF/G/H | adeL | adeI/J/K | adeN | adeN^ISAba1 | abaI | abaR | MEM | LEV | CAZ | PB | Biofilm | Motility |
|---|---|---|---|---|---|---|---|---|---|---|---|---|---|---|---|---|---|---|
| bla$_{OXA-23}$ | 1 | 0.676[a] | 0.796[a] | 0.195[b] | 0.539[a] | 0.208[b] | 0.132 | 0.188 | 0.042 | 0.403[a] | 0.233[b] | 0.573[a] | 0.746[a] | 0.656[a] | 0.717[a] | −0.023 | −0.418[a] | 0.219[b] |
| ISAba1 | | 1 | 0.739[a] | 0.300[a] | 0.619[a] | 0.222[b] | 0.196[b] | 0.103 | 0.056 | 0.332[a] | 0.192 | 0.581[a] | 0.737[a] | 0.568a | 0.711[a] | −0.011 | −0.348[a] | 0.085 |
| **Efflux systems** | | | | | | | | | | | | | | | | | | |
| adeA/B/C | | | 1 | 0.321[a] | 0.627[a] | 0.174 | 0.166 | 0.236[b] | 0.042 | 0.391[a] | 0.265[a] | 0.719[a] | 0.847[a] | 0.726[a] | 0.865[a] | 0.057 | −0.503[a] | 0.241[b] |
| adeS | | | | 1 | 0.443[a] | 0.003 | 0.202[b] | 0.109 | 0.069 | 0.269[a] | −0.025 | 0.128 | 0.318[a] | 0.217[b] | 0.314[a] | −0.032 | −0.002 | −0.049 |
| adeR | | | | | 1 | 0.143 | 0.223[b] | 0.317[a] | 0.046 | 0.235[b] | 0.491[a] | 0.459[a] | 0.625[a] | 0.524[a] | 0.609[a] | 0.058 | −0.277[a] | 0.105 |
| adeF/G/H | | | | | | 1 | 0.438[a] | 0.296[a] | 0.118 | 0.148 | −0.082 | 0.153 | 0.172 | 0.125 | 0.177 | 0.008 | −0.058 | 0.105 |
| adeL | | | | | | | 1 | 0.704[a] | −0.049 | 0.065 | −0.036 | 0.231[b] | 0.166 | 0.142 | 0.17 | 0.084 | −0.033 | 0.042 |
| adeI/J/K | | | | | | | | 1 | 0.109 | 0.092 | 0.168 | 0.328[a] | 0.236[b] | 0.203[b] | 0.241[b] | 0.119 | −0.14 | 0.143 |
| adeN | | | | | | | | | 1 | 0.322[a] | 0.051 | 0.128 | −0.019 | 0.034 | 0.071 | −0.082 | −0.12 | 0.007 |
| adeN^ISAba1 | | | | | | | | | | 1 | 0.04 | 0.281[b] | 0.400[a] | 0.325[a] | 0.349[a] | −0.083 | −0.200[b] | 0.171 |
| **Quorum sensing** | | | | | | | | | | | | | | | | | | |
| abaI | | | | | | | | | | | 1 | 0.346[a] | 0.263[a] | 0.187 | 0.253[b] | 0.185 | −0.18 | 0.027 |
| abaR | | | | | | | | | | | | 1 | 0.657[a] | 0.596[a] | 0.656[a] | 0.037 | −0.414[a] | 0.270[a] |
| **Antibiotic resistance** | | | | | | | | | | | | | | | | | | |
| MEM | | | | | | | | | | | | | 1 | 0.729[a] | 0.876[a] | −0.012 | −0.440[a] | 0.177 |
| LEV | | | | | | | | | | | | | | 1 | 0.824[a] | −0.107 | −0.409[b] | 0.219[b] |
| CAZ | | | | | | | | | | | | | | | 1 | −0.08 | −0.459[a] | 0.16 |
| PB | | | | | | | | | | | | | | | | 1 | −0.029 | 0.138 |
| Biofilm | | | | | | | | | | | | | | | | | 1 | −0.303[a] |
| Motility | | | | | | | | | | | | | | | | | | 1 |

[a] The correlation was significant at a confidence level of 0.01 (two-tailed).
[b] The correlation was significant at a confidence level of 0.05 (two-tailed).

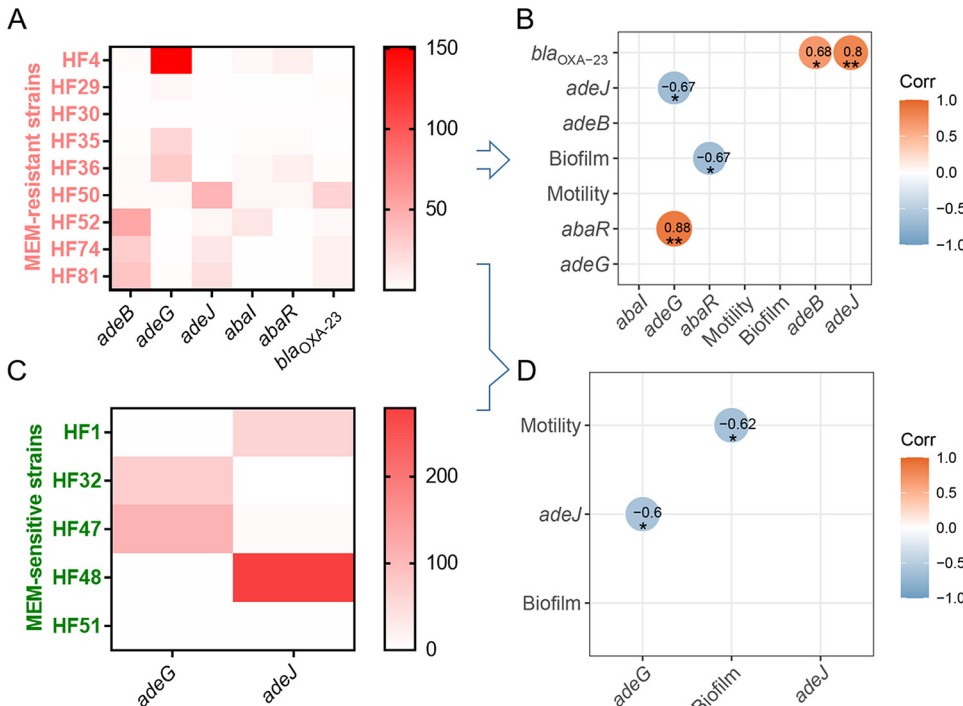

**FIG 2** Gene expression in representative strains of *A. baumannii*. (A, C) Heat maps of gene expression in representative MEM-resistant (A) and MEM-sensitive strains (C). The data are presented as means of three biological replicates. (B) Matrix graph of correlation (Corr) among gene expression, biofilm formation, and motility in representative MEM-resistant strains. (D) Matrix graph of correlation among gene expression, biofilm formation, and motility in representative MEM-resistant strains and MEM-sensitive strains. Arrows indicate that the representative strains used for correlation analysis were derived from strains used for gene expression heat map analysis. *, significantly correlated at a *P* value of 0.05; **, significantly correlated at a *P* value of 0.01.

antibiotics on the growth of planktonic cells varied on the strains and genotypes. A synergistic inhibition effect on the growth of planktonic cells of *A. baumannii* strains was observed in the combination of PB and MEM against HF15 (V type) and the combination of PB and CAZ against HF51 (IV-type) (Fig. 3). The additivity effect was observed in the combination of PB and LEV against HF1, HF15, and HF32 and in the combination

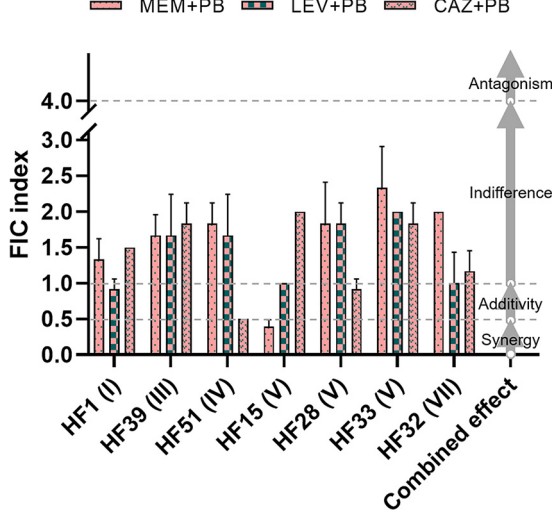

**FIG 3** Fractional inhibitory concentration (FIC) index of polymyxin B in combination with meropenem (MEM), levofloxacin (LEV), and ceftazidime (CAZ) against representative strains of *A. baumannii*. The combined antimicrobial effect was interpreted as synergy (FIC index [FICI] ≤ 0.5), additivity (FICI > 0.5 and <1), indifference (FICI ≥ 1 and < 4), and antagonism (FICI ≥ 4). The data are presented as means ± standard deviation (SD) of three biological replicates.

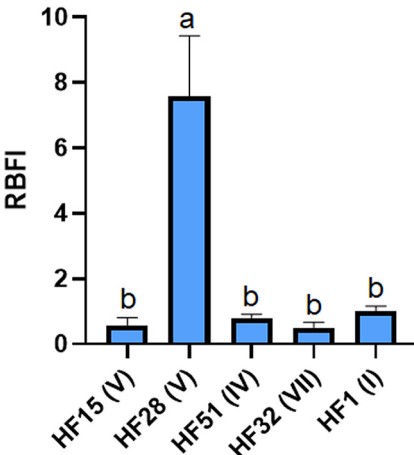

**FIG 4** Relative biofilm formatin index (RBFI) of representative strains of *A. baumannii*. The roman numerals (I to IX) indicate the different clades of strains identified from the ERIC-PCR-dependent fingerprints. The data are presented as means ± SD of three biological replicates and the significances were determined by nonparametric one-way analysis of variance (ANOVA). Columns with different lowercase letters (a and b) within treatments are significantly different at $P < 0.05$.

of PB and CAZ against HF28 (Fig. 3). The other combinations showed an indifference effect with FICs between 1 and 4 (Fig. 3). This observation is consistent with earlier studies (53, 54), indicating that PB combination therapy is potentially more effective against *A. baumannii* infections compared with monotherapy with PB. Strains of HF15, HF28, HF51, HF32, and HF1 that show FIC ≤ 1 (Fig. 3 and Fig. S4) to antibiotic combinations were further chosen to assess the antibiofilm properties.

**Robust biofilm is formed by an *A. baumannii* strain with genetic deficiency in AbaI/AbaR quorum sensing.** More than a role in MDR, RND-type efflux systems have been increasingly proven to be essential for biofilm formation by many Gram-negative bacteria (55). AbaI has been proven to nonspecifically produce a variety of quorum sensing molecules of *N*-acyl homoserine lactones (AHLs), including 3-OH-C12-HSL (56), that are likely to be the substrates for the RND-type efflux pumps and competitively bind with AbaR, leading to different responses in biofilm formation and motility (57). In this study, instead, the native expression of *abaI* (correlation [Corr] = −0.45; $P > 0.05$) and *abaR* (Corr = −0.67; $P < 0.05$) were both observed to be negatively correlated with the biofilm formation (Table 3). Meanwhile, the native expression of *adeG* that was positively correlated with *abaR* was observed to be negatively correlated with the expression of *adeJ* (Corr = −0.67; $P < 0.05$) and *adeB* (Corr = −0.52; $P > 0.05$) (Table 3), likely showing an AdeFGH-dependent AbaI/AbaR quorum sensing. However, *A. baumannii* does not seem to tilt the weight of quorum sensing toward biofilm formation under nonstress conditions, which is different from previous findings (22, 23). An inverse regulation between *adeG* and *adeJ* or *adeB* could be due to a fitness cost (58) during quorum sensing (Fig. 2). Moreover, typically as shown in Fig. 4, the strain HF28, which lacks the *abaR* gene (Fig. 1), exhibited a robust biofilm formation ability (RBFI = 7.57), while the strain HF32, which harbors a complete AbaI/AbaR system (Fig. 1), exhibited the lowest biofilm formation ability (RBFI = 0.51). In addition, an antibiotic-induced biofilm formation phenomenon was also observed in strains HF1, HF15, and HF51 (Fig. 5), all of which lack the *abaR* gene (Fig. 1). A deficiency in AbaI/AbaR quorum sensing was observed to be a common characteristic that was genetically displayed by IV and V type strains (Fig. 1; Table 2). These observations suggest that there are certain mechanisms other than AbaI/AbaR quorum sensing involved in the regulation of biofilm formation by *A. baumannii*. In addition, in view of the correlation of expression between *adeG* and *abaR*, it is also possible that AbaI/AbaR in turn regulates the expression of AdeFGH.

**Synergistic antibiofilm effects of PB in combination with CAZ against *A. baumannii*.** Typically, as shown in Fig. 5, the combined antibiofilm effects varied depending on the strains and the antibiotics used. Although the use of PB (1/8 × MIC,

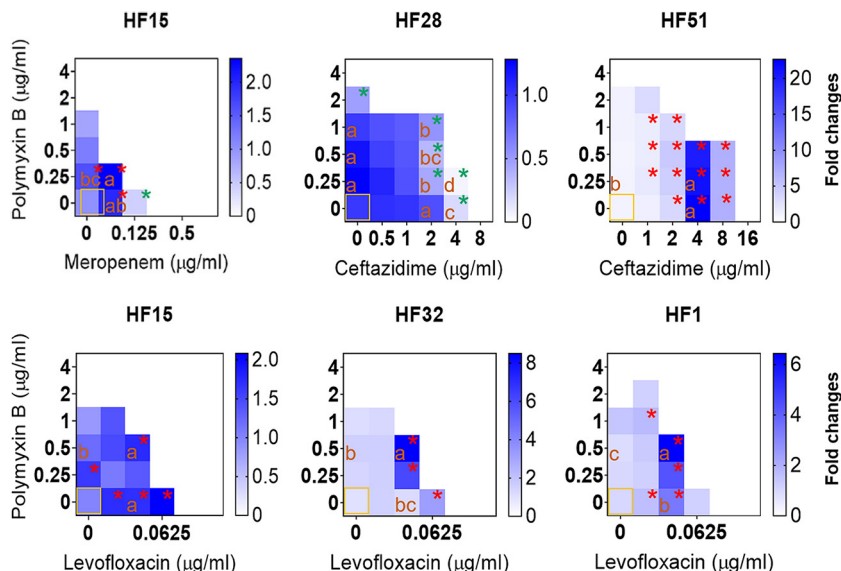

**FIG 5** Heat map of antibiofilm effects of polymyxin B in combination with meropenem, levofloxacin, and ceftazidime against representative strains of *A. baumannii*. Boxes with orange borders refer to control without antibiotic treatment. All data are presented as means of three biological replicates. The significances were determined by nonparametric one-way ANOVA. Red asterisks indicate significant increases at a *P* value of 0.05 compared to control. Green asterisks indicate significant decreases at a *P* value of 0.05 compared to control. Boxes with different lowercase (a to d) within treatments are significantly different at *P* < 0.05.

0.25 $\mu$g/mL) and MEM (1/4 × MIC, 0.0625 $\mu$g/mL) alone significantly increased the biofilm formation ability of HF15, the combined use of PB and MEM did not synergistically induce the biofilm formation under this subinhibitory concentration condition. Similarly, although LEV alone significantly increased the biofilm formation ability of HF15, the combined use of PB and LEV did not synergistically induce the biofilm formation. However, the combined use of PB (1/4 × MIC, 0.5 $\mu$g/mL) and LEV (1/4 × MIC, 0.03125 $\mu$g/mL) synergistically (*P* < 0.05) induced the biofilm formation of HF32 and HF1 compared to the individual induction effect. Particularly for HF28, a strain naturally having a robust biofilm formation ability (Fig. 4), although PB in combination with CAZ has failed to achieve a synergistic inhibitory effect on the growth of planktonic cells, the combination with a wide range of concentration of PB (from 1/4 to 1/16 × MIC) and CAZ (from 1/2 to1/4 × MIC), in contrast, exhibited a synergy inhibitory effect on the biofilm formation of this strain (Fig. 5). Induction of biofilm formation by *A. baumannii* has been found to be associated with the AbaI/AbaR quorum sensing in response to antibiotics (22, 23). Whether the achievement of the synergistic antibiofilm effect is related to a deficiency of AbaI/AbaR quorum sensing is worthy of further study, because it could be important for further consideration of its combined effect with quorum quenching (59). In addition, the biofilm formation of HF51 was significantly enhanced under the condition of a series of subinhibitory concentrations of CAZ alone, but PB and CAZ did not have a significant inducing effect on the formation of biofilm under almost all combinations of different concentrations (Fig. 5). Overall, compared with LEV and MEM, PB in combination with CAZ could be a suitable choice in terms of antibiofilm applications.

In summary, our data support and extend the analysis of the correlation among antibiotic resistance, biofilm formation, and pathogenicity. Carbapenem-sensitive *A. baumannii* strains were demonstrated to be less pathogenic but stronger to form biofilm. The major finding of this study is that the combination of PB and CAZ displays a significant synergistic inhibitory effect against robust biofilm formed by certain genotypes of *A. baumannii* strains that are characterized by having deficiency in AbaI/AbaR quorum

sensing. This study provides a theoretical basis for clinical applications of polymyxin-based combination in antibiofilm therapy.

## MATERIALS AND METHODS

**Bacterial collection and identification.** Bacterial samples were obtained from the First Affiliated Hospital of Anhui Medical University, a tertiary first-class hospital in China with 4,990 beds. A total of 103 *A. baumannii* strains were mainly collected from patients who were hospitalized in wards associated with intensive care unit (ICU), the respiratory and critical illness unit, and the burn unit from July to October 2020 (Table S1). The identification of these *A. baumannii* strains was performed with matrix-assisted laser desorption ionization-time of flight mass spectrometry (MALDI-TOF MS) on a Vitek MS system (bioMérieux, Marcy l'Étoile, France).

**MIC determination assay.** The susceptibilities to meropenem (MEM), levofloxacin (LEV), ceftazidime (CAZ), and polymyxin B (PB) of *A. baumannii* strains were tested in 103 *A. baumannii* strains according to a previously used broth microdilution method (22) with some modification. Briefly, culture medium with or without bacterial cells was served as positive and negative controls, respectively. The optical density (OD) of each well was determined at 600 nm in a BioTek Synergy2 microplate reader. The inhibition rate (%) was calculated as $[(OD_{positive\ control} - OD_{negative\ control}) - (OD_{treatment} - OD_{negative\ control})]/(OD_{positive\ control} - OD_{negative\ control}) \times 100\%$. The MIC was determined as the lowest concentration of antibiotic at which there was growth inhibition of $\geq$ 99%. The MIC values obtained by using cation-adjusted Mueller-Hinton (MH) medium and nutrient broth (NB) medium, respectively, were compared in 103 clinical strains and a type strain of *A. baumannii* ATCC 19606. The susceptibilities of at least three independent replicates were tested and interpreted according to CLSI guidelines.

**Biofilm formation assay.** The biofilm formation assays were performed according a previous method (22) with slight modification. Briefly, the biofilm remaining in each well was stained with a 0.1% crystal violet (CV) (Sigma) solution at 37°C for 10 min. The stained biofilm cells were destained with 95% ethanol and measured at 600 nm ($CV_{biofilm}$). The wells without inoculation were stained and used as the negative control ($CV_{control}$) to reduce the background staining from the CV-stained biofilm cells. The ability to form a biofilm was expressed using a biofilm formation index (BFI = $[CV_{biofilm} - CV_{control}]/OD_{planktonic}$). The type strain *A. baumannii* ATCC 19606 was used as a positive standard strain for biofilm formation to calculate the relative biofilm formation capacity (RBFI = $BFI_{isolate}/BFI_{A.\ baumannii\ ATCC\ 19606}$). The biofilm formation of at least three independent replicates for 103 clinic *A. baumannii* strains was tested.

**Surface motility assay.** Surface motility assay was carried out by using Luria-Bertani media (Difco) solidified with 0.5% agar in petri dishes. The dishes were point-inoculated from an active culture with a sterile iron wire and incubated at 37°C for 24 h. Swarming motility was assessed by measuring the circular turbid zones formed by the bacterial cells migrating away from the point of inoculation. The surface motility was detected in at least three independent replicates for 103 clinic *A. baumannii* strains.

**Gene detection.** A total of 103 *A. baumannii* strains were detected for the presence of genes, including RND efflux-associated *adeA*, *adeB*, *adeC*, *adeS*, *adeR*, *adeF*, *adeG*, *adeH*, *adeL*, *adeI*, *adeJ*, *adeK*, and *adeN*; quorum sensing-associated *abaI* and *abaR*; carbapenemase *bla*_OXA-23_; and insertion sequence IS*Aba1* by PCR method with the specific primers showed in Table S2. DNA was extracted with QIAamp DNA blood kit (Qiagen, Valencia, CA). The amplification conditions were 94°C for 5 min; 35 cycles of 30 s at 94°C, 30 s at 56°C, and 2 min at 72°C; and a final extension of 7 min at 72°C. Amplicons were electrophoresed in gel agarose 1.5% containing 0.1% GoldView and visualized under gel documentation system with UV transilluminator (Bio-Rad). Each strain was detected in at least two independent replicates to confirm the presence of genes.

**Molecular fingerprinting.** The enterobacterial repetitive intergenic consensus (ERIC) sequence-based PCR fingerprinting assay was performed with a previously used method (22). Fingerprints were determined on the basis of at least three independent replicates of tests. A hierarchical cluster analysis was performed using the average linkage method described by Nei and Li (60). Cytoscape software was used to generate the network between genotypes and bacterial source.

**Reverse Transcription-PCR assays.** Transcriptional analysis was performed according to a previously used method (22) and with a minor modifications. Briefly, total RNA was extracted with the RNeasy Protect Bacteria minikit (Qiagen). cDNA was synthesized using a reverse transcription kit (Qiagen) in a 20-$\mu$l reaction system. The PCR amplification was performed using the ABI 7500 real-time PCR system (Applied Biosystems, Foster City, CA). Nine carbapenem-resistant strains and five carbapenem-sensitive strains were randomly chosen for transcriptional analysis. Each sample was run in triplicate. The comparative threshold cycle ($C_T$) method was used to analyze the relative expression of the targeted genes. The $C_T$ value of the reference gene *rpoB* was used to calculate the $\Delta C_T$.

**Antimicrobial checkerboard assays for planktonic *A. baumannii*.** Since the MIC value of highly resistant strains cannot be detected within the limited concentration range of antibiotics, drug-sensitive strains or low-resistant strains were considered for checkerboard assays. Finally, seven representative strains (four representative strains with I, III, IV, and VII type, respectively, and three representative strains including HF28 [the most robust strains in biofilm formation] with V type) were chosen to investigate the combined effects of antibiotics against strains with different genotypes. A broth microdilution checkerboard procedure (61) with a slight modification was used to determine the inhibitory effect of antibiotic combinations on the growth of planktonic cells of *A. baumannii* strains. Each drug was serially diluted at six concentrations to create an 6 $\times$ 6 matrix in a 96-well plate. Briefly, antibiotic A (PB) was diluted in wells along the abscissa on the left half of the plate, while antibiotic B (MEM, LEV, or CAZ) was diluted in wells along the ordinate on the right half of the plate. The dilutions of antibiotic B were then

parallelly transferred and mixed with antibiotic A at the left half plate correspondingly. After inoculation, the plates with a bacterial population of approximately $10^6$ CFU/mL in final 200 $\mu$l of each well were incubated with at 37°C for 20 h. Each test was performed in three independent replicates. Absorbance at 600 nm of bacterial culture was collected after incubation. The fractional inhibitory concentration (FIC) was calculated for the first well with bacterial growth inhibition ≥99% in each row of the micro-plate containing all antimicrobial agents as follows: FIC of drug A ($FIC_A$) = MIC of drug A in combination/ MIC of drug A alone, and FIC of drug B ($FIC_B$) = MIC of drug B in combination/MIC of drug B alone. According to method described by Lewis et al. (62), the combined antimicrobial effect was evaluated by using a FIC index (FICI) = $FIC_A$ + $FIC_B$ and interpreted as synergy (FICI ≤ 0.5), additivity (0.5 < FICI < 1), indifference (1 ≤ FICI < 4), and antagonism (4 ≤ FICI). The FICs were confirmed based on three independent replicates of test.

**Antibiofilm checkerboard assays.** Five representative strains with FIC ≤ 1 in response to different antibiotic combinations were further tested to observe the combined effects of antibiotics on the biofilm formation. This assay was evaluated on the basis of the modified two-dimensional checkerboard method and the biofilm formation assay described above. Briefly, the well at the top right corner of the 6 × 6 matrix was chosen as the negative control to reduce the background staining from the CV-stained biofilm cells. Biofilm formed in the well at the bottom left corner of the matrix was used as the positive control to calculate the fold changes in biofilm formation. The biofilm having the most significant change in formation in the combination matrix was vertically and horizontally compared with that formed under conditions of exposure to the corresponding concentration of each single antibiotic. Synergistic effect in biofilm formation between two antibiotics was defined as a combined change in value that is statistically greater than the sum of the individual changes. The data were collected based on three independent replicates of tests.

**Data analysis.** The experimental data were analyzed using IBM SPSS Statistics 19. The chi-square test was used for the association of categorical variables, while Student's $t$ test was used for the comparison of means and the correlation of variables used Spearman's correlation coefficient, with $\rho$ between −1 and +1.

## SUPPLEMENTAL MATERIAL

Supplemental material is available online only.

**SUPPLEMENTAL FILE 1**, PDF file, 0.8 MB.

## ACKNOWLEDGMENTS

This research was supported by National Nature Science Foundation of China grant 82072297 and Open Project Program of Jiangsu Key Laboratory of Zoonosis grant R1908.

Y.L. worked on the biofilm formation, motility, and gene expression, and checkerboard test; B.W. contributed to bacterial isolation, identification, antimicrobial susceptibility test, and manuscript modification; F.L. and J.A. revised the manuscript; W.Z. was involved in MIC detection, checkerboard test, and gene expression; L.C., J.X., Y.Y., Q.C., and Z.R. were involved in data analysis. X.H. analyzed the data, drafted, and revised the manuscript. All authors read and approved the final manuscript.

We declare no conflict of interest.

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
