## [Reviewer comments · Microbiology Spectrum]

Microbiology Spectrum

Synergistic inhibitory effect of polymyxin B in combination with caftazidime against robust biofilm formed by *Acinetobacter baumannii* with genetic deficiency in Abal/AbaR quorum sensing

Yinyue Li, Bo Wang, Feng Lu, Juhee Ahn, Wenwen Zhang, Liangliang Cai, Jiahui Xu, Yi Yin, Qingchao Cao, Zhenyu Ren, and Xinlong He

Corresponding Author(s): Xinlong He, Yangzhou University

Review Timeline:

Submission Date:	October 2, 2021
Editorial Decision:	December 4, 2021
Revision Received:	January 6, 2022
Editorial Decision:	January 21, 2022
Revision Received:	January 23, 2022
Accepted:	January 28, 2022

Editor: William Lainhart

Reviewer(s): Disclosure of reviewer identity is with reference to reviewer comments included in decision letter(s). The following individuals involved in review of your submission have agreed to reveal their identity: Erin S Gloag (Reviewer #2)

Transaction Report:

DOI: <https://doi.org/10.1128/spectrum.01768-21>

December 4, 2021

Dr. Xinlong He
Yangzhou University
Department of Pathogen Biology, School of Medicine
Yangzhou, Jiangsu 225009
China

Re: Spectrum01768-21 (Balanced evolution of carbapenem resistance versus biofilm formation, and synergistic anti-biofilm effect of polymyxin B in combination with caftazidime against *Acinetobacter baumannii*)

Dear Dr. Xinlong He:

Link Not Available

Sincerely,

William Lainhart

Journals Department
Reviewer comments:

Reviewer #1 (Comments for the Author):

The most interesting aspect of this paper is the effect of antibiotic combinations on biofilm formation and that is where the focus should be. It is possible that organisms that are trying to survive antibiotic exposure resort to biofilm formation as a form of protection. The incubation is not long enough to really establish a protective biofilm, but this work could explain what conditions in terms of drug exposure promote biofilm formation. Much of the manuscript involves genotyping and detection of resistance-related genes that is not novel.

Line 87 -The authors talk about polymyxin combination therapy and reduction in nephrotoxicity. This is only true if the dose of polymyxin is reduced. I would recommend that that this statement be removed to further focus the article.

Line 175 - Does the spectrophotometric growth inhibition of >99% correlate with visual interpretation? Just want to know if there is any systematic difference in this reading method and the standard visual reading of broth microdilution testing.

Line 178 - For FIC index there is usually some allowance for technical variation where synergy is defined as FIC index {less than or equal to} 0.25 and antagonism is defined as {greater than or equal to} 4. It is difficult to make any interpretations of the

interaction with FIC index of 0.5 as synergy as this could be sporadic and not reproducible.

Line 182 - Referring to the upper right corner and bottom left of the matrix is not a clear description. Are you using a 8 x 12 well plate? I would refer to abscissa as the 8 rows (A-H) and the ordinate as the 12 columns (1-12). Are you using a 8x8 grid for dilutions? The right half and left half implies that you are using all 12 columns.

Line 203 -The manuscript would be easier to read if too many significant figures were avoided. For some of the percent values are 2, 3, or 4 significant digits with no apparent difference in precision. Three should be adequate.

Statistics - There are several uses of the Spearman Rank Correlation test. Given the data involved this simplifies to a 2 X 2 Chi square test. I wonder if readers would more likely understand the application of a chi square test.

Line 248 - The correlation between the two fingerprinting methods is not intuitive. ERIC PCR was completed and allowed clustering of the organisms into 5 groups. Separately, the organism were subjected to CRGV-PCR, and organisms were clustered into 6 groups. Since the assay is looking at different genes why would the cluster designation be correlated between the methods. This analysis would also indicate some sort of hierarchy to the order of clusters and some reason to suspect that the cluster hierarchy is expected to be related in some way. Aren't these two methods distinct ways of grouping organisms?

Line 340 - Defining an FIC index of 0.5 as synergy is problematic as the result would not be reproducible. Although, this would indicate synergy on a strictly theoretical basis, when one accounts for doing two-fold dilutions and minor error, it is not conceivable to expect additivity to always result as 1. Rather, a range of 0.5 to 2 would be expected. Also, this would be very inconsistent with the way the drugs are used. For CRAB isolates with higher MIC values for meropenem, achieving a concentration of 1/2 MIC is hard to sustain due to dosing limitations.

Line 357 "This is again proof of a finding as mentioned above that biofilm formation more virulent *A. baumannii* strains tend to be more susceptible to certain antibiotics". Is the intent to say that virulence and ability to form biofilm are related? I don't think that biofilm formation necessarily equates to virulence. Biofilm associated infections tend to involve foreign bodies and result in more of a chronic infection issue.

Overexpression of blaOXA51 is associated with ISAba1 insertion gene. Oxa-51 was not assessed in this study; however, given that it is a major carbapenemase, the limitations of not detecting this beta-lactamase should be discussed.

Figures 1, 2, 3 - could be moved to supplementary materials.

Figure 4 really does not add any useful information. I would conclude that there is no difference in RSFI and surface mobility based on ERIC-type or MEM susceptibility.

Figure 6 -The heat map panels for FIC index are not clear to me. The "inhibition" is graded from 0 to 100, but what is meant by inhibition. I gather that these are 6 different strains of *A. baumannii*. It would be important to know if RBF1 within an isolate is related to FIC yet the X and Y axis for panel D appears to be raw concentration. For some of those isolates, it appears that there are very narrow conditions required to promote biofilm formation. If inhibition is achieved, biofilm is suppressed. However, antibiotics at marginal subinhibitory concentrations promote the most biofilm formation.

Reviewer #2 (Comments for the Author):

Here authors analyse the drug resistant phenotypes of 103 clinical isolates of *Acinetobacter baumannii* and correlated these phenotypes to the presence of other genes and biofilm formation and motility. This is an interesting manuscript to read, and authors have performed an impressive amount of work. My main critique is that overall, the manuscript is lacking detail, particularly in the methods and description of the figures. My specific comments are below.

Major comments

1. Not all the figures or figure panels are referred to in the text.
2. There is insufficient detail provided in the methods section. For example
 - a. 'Bacterial collection and identification'. Please include a supplemental table listing the 103 strains. Include information on the source of each strain and any known genetic markers.
 - b. L126. What was the negative control?
 - c. L133. Was the media solidified with 0.5% agar? Or was the media diluted to 0.5%? This is not clear.
 - d. Indicate the number of biological and technical replicates for each assay.
 - e. L141 - 142. No DNA extraction method is indicated.
 - f. L149. What does the abbreviation CRGV-PCR refer to? Has this method been published before? If so include a reference. If not provide details of this fingerprinting method.
 - g. L170. Was 200 uL the final volume in the wells or the inoculum volume?
 - h. L174 - 181. Provide a reference for the FIC calculation. Also in more recent publications I have seen the additive range as 0.5 - 4, and antagonism > 4.

3. Figure 1. It is difficult to differentiate the black and grey lines in the figures. Provide more detail in the figure legend and mention that the roman numerals indicate the different clades identified from the fingerprinting.
4. L242. What is the difference between groups and strains in the context?
5. Figure 2. Provide more information in the figure legend. What does the double headed arrow mean? Font size is too small for axis and labels.
6. L246 - 247. Statistics is not indicated on the figure.
7. Figure 5. Numbers on the plots are too small. What does the bracket indicate? Provide more detail in the figure legend.
8. L281-286. This section is unclear.
9. Figure 3. Provide more detail in the legend. Text in the figure is too small.
10. L301 - 306. This section is unclear. Motility is usually downregulated during biofilm formation.
11. L338 - 341. This does not seem to be supported by Fig 6A, as the majority seem to have $FIC > 1$. Please clarify.
12. L341. What was the rationale for selecting these strains?
13. L357 - 359 and L365 - 366. This is an over interpretation.
14. Figure 4 is not referenced in the text. Provide more detail in the figure legend, there is no mention of the individual panels. What do the arrows mean?
15. Figure 6. Provide a rational for why these specific strains were selected. Why are heat maps only shown for select antibiotic combinations for select strains? Figure 6A - assuming this is for planktonic cells? Provide more detail in the figure legend.

Minor comments

1. The title is rather long, and is almost a combination of two titles. I would suggest coming up in a new title that is shorter and conveys a single point.
2. L28. In combination with what?
3. L29. What do authors mean by 'potency' in this context?
4. L31 and 34. What are these abbreviations of?
5. L85 - 87. Mix of past and present tenses in this sentence.
6. L199. Co-evolved?
7. Fig 2D. Font size for the values above the bars and the key are too small.
8. L214. 'Eastern'
9. Table 2 is referenced in the text before Table 1.
10. L254 - 255. I do not understand this title.
11. L260, 267, 297. What do authors mean by 'synchronization'? Maybe replace with 'correlation'?
12. L261 - 263. What do authors mean by 'nepotism' in this context?
13. Renumber figures in the order they are referenced in the text.
14. L299. What do authors mean by 'contamination' in this context?
15. L312, 337. Change to 'Gram-negative'.
16. L310 - 311. This title doesn't make sense. Maybe some words missing?
17. L364 - 365. What do authors mean by 'partial genotypes'?

Staff Comments:

Preparing Revision Guidelines

Please return the manuscript within 60 days; if you cannot complete the modification within this time period, please contact me. If you do not wish to modify the manuscript and prefer to submit it to another journal, please notify me of your decision immediately so that the manuscript may be formally withdrawn from consideration by Microbiology Spectrum.

Responses to Reviewers

Re: Spectrum01768-21 (Balanced evolution of carbapenem resistance versus biofilm formation, and synergistic anti-biofilm effect of polymyxin B in combination with caftazidime against *Acinetobacter baumannii*)
Authors: Yinyue Li, *et al.*

We sincerely thank all Reviewers for your valuable comments and giving us the chance to revise our manuscript. We have finished the point-by-point responses for this manuscript, and you will find the changes in the Marked Up Manuscript up-loaded.

Responses to Reviewer 1

Line 87 -The authors talk about polymyxin combination therapy and reduction in nephrotoxicity. This is only true if the dose of polymyxin is reduced. I would recommend that that this statement be removed to further focus the article.

→ As suggested, this statement has been removed.

Line 175 - Does the spectrophotometric growth inhibition of >99% correlate with visual interpretation? Just want to know if there is any systematic difference in this reading method and the standard visual reading of broth microdilution testing.

→ Using spectrophotometric growth inhibition method to determine the MIC value has been referred in a certain number of publications (reference DOIs: 10.1038/s41467-018-02875-z and 10.1128/AAC.00877-15). We found there was no obvious difference between the two methods for the determination of MIC values. However, in some cases, when visual reading was used, different people had inconsistencies in the identification of the first well with no obvious growth of bacteria cell. Using spectrophotometric growth inhibition method could avoid the difference caused by subjective factors.

Line 178 - For FIC index there is usually some allowance for technical variation where synergy is defined as FIC index {less than or equal to} 0.25 and antagonism is defined as {greater than or equal to} 4. It is difficult to make any interpretations of the interaction with FIC index of 0.5 as synergy as this could be sporadic and not reproducible.

→ Indeed, there is controversy about the determination of the effect of combination drugs. We agree that FIC index of 0.25 is more strict in the determination of synergistic effect. However, it is difficult to find references that use 0.25 as the threshold for synergy, and it is common to see 0.5 as a threshold for judging synergistic effect in publications (DOIs: 10.1093/jac/49.2.345, 10.1093/jac/dkg301, 10.1002/advs.201902227, etc.). Fortunately, no matter which judgment standard is adopted, it will not affect the determination of the result of anti-biofilm effect. After considering the suggestions of all reviewers, finally, in this study, with reference to the publication (DOI: 10.1093/jac/49.2.345), the synergy was temporarily defined as an $FIC \leq 0.5$, and additivity was defined as > 0.5 and < 1 . Indifference was defined as an $FIC \geq 1$ and < 4 , whereas antagonism was defined as an FIC of ≥ 4 . We hereby declare that we do not rule out considering 0.25 as the threshold of synergy if we find relevant references. The reference has been provided in this manuscript (Line 195-196). Relevant changes have also been made at Figure 3. (Line 674)

Line 182 - Referring to the upper right corner and bottom left of the matrix is not a clear description. Are you using a 8 x 12 well plate? I would refer to abscissa as the 8 rows (A-H) and the ordinate as the 12 columns (1-12). Are you using a 8x8 grid for dilutions? The right half and left half implies that you are using all 12 columns.

→ We agree that the description of the upper right corner and bottom left of the matrix is not clearly understood if we did not mention the size of matrix in advance. For checkerboard assay, actually we created a 6 × 6 matrix in a 8 × 12 well plate. Compared with 8 × 8 matrix, the advantage of 6 × 6 matrix is that the dilution preparation of the two drugs can be done on the same plate. For easy to understand, relevant changes have been made at method part. (Line 182-183, 204-205)

Line 203 -The manuscript would be easier to read if too many significant figures were avoided. For some of the percent values are 2, 3, or 4 significant digits with no apparent difference in precision. Three should be adequate.

→ As suggested, the significant digit of all percent values has been changed to 3 in the text.

Statistics - There are several uses of the Spearman Rank Correlation test. Given the data involved this simplifies to a 2 X 2 Chi square test. I wonder if readers would more likely understand the application of a chi square test.

→ As suggested, a chi-square test was applied in data analysis, and relevant changes have been made at data analysis part. (Line 214)

Line 248 - The correlation between the two fingerprinting methods is not intuitive. ERIC PCR was completed and allowed clustering of the organisms into 5 groups. Separately, the organism were subjected to CRGV-PCR, and organisms were clustered into 6 groups. Since the assay is looking at different genes why would the cluster designation be correlated between the methods. This analysis would also indicate some sort of hierarchy to the order of clusters and some reason to suspect that the cluster hierarchy is expected to be related in some way. Aren't these two methods distinct ways of grouping organisms?

→ ERIC-PCR fingerprinting has been demonstrated to be a reliable technique for discriminating intraspecific variations. CRGV-PCR method attempts to analyze the genetic differences in terms of the distribution of 17 genes among 103 strains. The original intention of the correlation analysis between ERIC-PCR and CRGV-PCR was to illustrate the correlation between intraspecific variations and the distribution of 17 genes. Given the fact that this method of analysis is difficult to understand and has never been used before, it has been abandoned in this study. Instead, a direct description of the distribution of genes in different genotype strains was used. Relative changes have been made at Line 219-266, 655 (Figure 1), 711 (Figure S2).

Line 340 - Defining an FIC index of 0.5 as synergy is problematic as the result would not be reproducible. Although, this would indicate synergy on a strictly theoretical basis, when one accounts for doing two-fold dilutions and minor error, it is not conceivable to expect additivity to always result as 1. Rather, a range of 0.5 to 2 would be expected. Also, this would be very inconsistent with the way the drugs are used. For CRAB isolates with higher MIC values for meropenem, achieving a concentration of 1/2 MIC is hard to sustain due to dosing limitations.

→ We completely agree with the reviewer's point of view. However, as we mentioned above, it is difficult to find references with 0.25 as the boundary of synergy, and the citation of references is required. Therefore, here, we have to adopt most of the standards cited in the literature, taking 0.5 as the demarcation point for synergistic effects, 1 as the demarcation point for additive effects, and 4 as the demarcation point for antagonistic effects.

Indeed, for isolates with higher MIC values for antibiotics including meropenem, achieving a concentration of 1/2 MIC is hard to sustain due to dosing limitations. Therefore, the representative strains used in the checkerboard test were susceptible strains or strains with lower levels of resistance to these antibiotics. A statement has been provided. (L173-180)

Line 357 "This is again proof of a finding as mentioned above that biofilm formation more virulent *A. baumannii* strains tend to be more susceptible to certain antibiotics". Is the intent to say that virulence and ability to form biofilm are related? I don't think that biofilm formation necessarily equates to virulence. Biofilm associated infections tend to involve foreign bodies and result in more of a chronic infection issue.

→ We agree with the reviewer's point of view. This sentence has been deleted.

Relevant descriptions have been changed (L81-85, 318-320).

Overexpression of blaOXA51 is associated with ISAbal insertion gene. Oxa-51 was not assessed in this study; however, given that it is a major carbapenemase, the limitations of not detecting this beta-lactamase should be discussed.

→ As suggested, the limitation of not taking this beta-lactamase into consideration has been discussed. (L282-285)

Figures 1, 2, 3 - could be moved to supplementary materials.

→ Figures 1 and 2 have been made some modification. As suggested, Figures 1 and 3 have been changed to Figure S2 and Figure S3 as supplementary materials (L711 and 722), original Figure 2 has been modified and changed to current Figure 1 (L655).

Figure 4 really does not add any useful information. I would conclude that there is no difference in RSFI and surface mobility based on ERIC-type or MEM susceptibility.

→ Indeed, data analysis indicates that there is no difference in RBFi and surface mobility based on ERIC-type. As suggested, original Figure 4 has been deleted, and changes of relevant statement has been made at L323-325.

Figure 6 -The heat map panels for FIC index are not clear to me. The "inhibition" is graded from 0 to 100, but what is meant by inhibition. I gather that these are 6 different strains of *A. baumannii*. It would be important to know if RBFi within an isolate is related to FIC yet the X and Y axis for panel D appears to be raw concentration. For some of those isolates, it appears that there are very narrow conditions required to promote biofilm formation. If inhibition is achieved, biofilm is suppressed. However, antibiotics at marginal subinhibitory concentrations promote the most biofilm formation.

→ "inhibition" means the inhibition of the growth of planktonic cells.

Since the MIC value of highly resistant strains cannot be detected within the limited concentration range of antibiotics, drug-sensitive strains or low-resistant strains are considered objects for combined drug experiments. In order to understand the combined effects of antibiotics against different genotype strains and different strains of the same genotype, finally 7 representative strains (4 representative strains with I, III, IV, and VII type, respectively, and 3 representative strains including HF28 (the most robust strains in biofilm formation) with V type) were chosen for the checkerboard assays.

FIC index (original Figure 6-A) showed the fractional inhibitory concentration of antibiotic combination against the growth of planktonic cells of these 7 strains. In this study, certain combinations of antibiotics showed a synergistic or indifference effect on the growth of planktonic cells of 7 strains, and the original Figure 6-B showed a concrete presentation of the checkerboard experiment of these 5 strains. Then, we further tested and compared the intrinsic biofilm formation (without antibiotic treatment) of 5 strains (HF15, HF28, HF51, HF32, HF1) with $FIC \leq 1$ (original Figure 6-C), and the inducibility of biofilm formed by these 5 strains under conditions of antibiotic combinations (original Figure 6-D). The biofilm formation by those strains that showing $FIC > 1$ was not further observed in this study.

Descriptions have been made to distinguish the results of the checkerboard experiment on planktonic bacteria cells and biofilm. (L182, 342-343, 346-353)

The statement for strain selecting has been provided. (L173-180, 200-202, 354-356)

Reviewer #2 (Comments for the Author):

Here authors analyse the drug resistant phenotypes of 103 clinical isolates of *Acinetobacter baumannii* and correlated these phenotypes to the presence of other genes and biofilm formation and motility. This is an interesting manuscript to read, and authors have performed an impressive amount of work. My main critique is that overall, the manuscript is lacking detail, particularly in the methods and description of the figures. My specific comments are below.

Major comments

1. Not all the figures or figure panels are referred to in the text.
→ The figures without mentioned in text and invalid figures including original Figure 4 that show no significant differences between groups have all been deleted.
2. There is insufficient detail provided in the methods section. For example

a. 'Bacterial collection and identification'. Please include a supplemental table listing the 103 strains. Include information on the source of each strain and any known genetic markers.

→ As suggested, a table listing the information on the source of each strain and ERIC-PCR-dependent genetic types has been made as supplementary materials. (Table S1, L697)

b. L126. What was the negative control?

→ The negative control refers those experimental groups (or wells) without inoculation. A clear description has been made at (L131-132).

c. L133. Was the media solidified with 0.5% agar? Or was the media diluted to 0.5%? This is not clear.

→ A clear description has been made at (L139-140).

d. Indicate the number of biological and technical replicates for each assay.

→ As suggested, the number of biological and technical replicates for each assay have been provided in MATERIALS AND METHODS part.

e. L141 - 142. No DNA extraction method is indicated.

→ As suggested, DNA extraction method has been provided (L150-151).

f. L149. What does the abbreviation CRGV-PCR refer to? Has this method been published before? If so include a reference. If not provide details of this fingerprinting method.

→ CRGV-PCR refers the carbapenem resistance associated genetic variation in terms of the presence or absence of 17 genes related to RND-type efflux, AbaI/AbaR quorum sensing, carbapenemase OXA-23, and ISAbaI element. It has not been published.

CRGV-PCR method attempts to analyze the genetic differences in terms of the distribution of 17 genes among 103 strains. The original intention of the correlation analysis between ERIC-PCR and CRGV-PCR was to illustrate the correlation between intraspecific variations and the distribution of 17 genes. Given the fact that this method of analysis is difficult to understand and has never been used before, it

has been abandoned in this study. Instead, a direct description of the distribution of genes in different genotype strains was used. Relative changes have been made at Line 219-266, 655 (Figure 1), 711 (Figure S2).

g. L170. Was 200 uL the final volume in the wells or the inoculum volume?

→ It refers to the final volume in the wells. For more clear description, this sentence has been changed. (L187-189)

h. L174 - 181. Provide a reference for the FIC calculation. Also in more recent publications I have seen the additive range as 0.5 - 4, and antagonism > 4.

→ As suggested, a reference has been provided. The combined antimicrobial effect was interpreted as synergy ($FICI \leq 0.5$), additivity ($FICI > 0.5$ and < 1), indifference ($FICI \geq 1$ and < 4), and antagonism ($FICI \geq 4$). (L195-196)

3. Figure 1. It is difficult to differentiate the black and grey lines in the figures.

Provide more detail in the figure legend and mention that the roman numerals indicate the different clades identified from the fingerprinting.

→ As suggested, the figure has been modified, and the description of roman numerals has been provided. This figure has been moved to supplementary materials (Figure S2, L711).

4. L242. What is the difference between groups and strains in the context?

→ As suggested, a description of group and strains has been made in the context (L227-229).

5. Figure 2. Provide more information in the figure legend. What does the double headed arrow mean? Font size is too small for axis and labels.

→ As suggested, this figure has been modified and numbered as Figure 1 in revised manuscript, and a detail description has been provided in figure legend. To avoid misunderstanding, the double headed arrow has been deleted in modified figure. Font size of labels has been enlarged. (L655, Figure 1)

6. L246 - 247. Statistics is not indicated on the figure.

→ Considering that the CRGV-PCR typing method has been abandoned in this study, this problem will no longer exist.

7. Figure 5. Numbers on the plots are too small. What does the bracket indicate?

Provide more detail in the figure legend.

→ This figure has been modified and numbered as Figure 2 in revised manuscript. Font size of numbers has been enlarged. As suggested, a detail description has been provided in figure legend. (L666, Figure 2)

8. L281-286. This section is unclear.

→ The sentence has been changed for easy understanding. (L329-331)

9. Figure 3. Provide more detail in the legend. Text in the figure is too small.

→ As suggested, this figure has been modified and numbered as Figure S3 as supplementary materials. (Figure S3, L722-726)

10. L301 - 306. This section is unclear. Motility is usually downregulated during biofilm formation.

→ These sentence has been modified for easy understanding. (L333-338)

11. L338 - 341. This does not seem to be supported by Fig 6A, as the majority seem to have $FIC > 1$. Please clarify.

→ As suggested, a corrected description has been made. (L346-351)

12. L341. What was the rationale for selecting these strains?

→ The rationale for selecting these strains has been described. (L173-180, 200-202, 354-356)

13. L357 - 359 and L365 - 366. This is an over interpretation.

→ These sentence have been deleted in the context.

14. Figure 4 is not referenced in the text. Provide more detail in the figure legend, there is no mention of the individual panels. What do the arrows mean?

→ Considering original Figure 4 showed no significant differences between groups, it has been deleted in revised manuscript.

15. Figure 6. Provide a rational for why these specific strains were selected. Why are heat maps only shown for select antibiotic combinations for select strains? Figure 6A - assuming this is for planktonic cells? Provide more detail in the figure legend.

→ As suggested, the rationale for selecting these strains has been described in the method part. The statement for strain selecting has been provided. (L173-180, 200-202, 354-356)

→ Original Figure 6A showed the antimicrobial effect for planktonic cells. This Figure has been modified and number as Figure 3. More detail description has been provided in figure legend. (Figure 3, L674)

Minor comments

1. The title is rather long, and is almost a combination of two titles. I would suggest coming up in a new title that is shorter and conveys a single point.

→ As suggested, the title has been changed. (L1-3)

2. L28. In combination with what?

→ A complete sentence has been provided. (L29)

3. L29. What do authors mean by 'potency' in this context?

→ 'potency' has been changed to 'capacity'. (L30)

4. L31 and 34. What are these abbreviations of?

→ Full name or a detail description has been provided for ERIC-PCR, RND, AbaI/AbaR, OXA, and ISAba1. (L31-35)

5. L85 - 87. Mix of past and present tenses in this sentence.

→ The present tense has been changed to past tense. (L92)

6. L199. Co-evolved?

→ The sentence has been changed, 'co-evolved' has been deleted. (L219-220)

7. Fig 2D. Font size for the values above the bars and the key are too small.

→ Fig 2D has been changed to Table 1. (L643)

8. L214. 'Eastern'

→ It has been changed. (L248)

9. Table 2 is referenced in the text before Table 1.

→ The serial number of all tables has been adjusted according to the order they appeared in the text.

10. L254 - 255. I do not understand this title.

→ The title has been changed. (L267-268)

11. L260, 267, 297. What do authors mean by 'synchronization'? Maybe replace with 'correlation'?

→ As suggested, 'synchronization' has been replaced with 'correlation'. For easy understanding, 'synchronized' appeared in text has also been replaced with 'correlated'. (L274, 280, 308, 340, 383)

12. L261 - 263. What do authors mean by 'nepotism' in this context?

→ 'nepotism' was originally used to express the neighboring relationship between two genes in structure. For easy understanding, 'which form a kind of nepotism' has been changed to 'which forms a neighbor relation in genome structure' in sentence. (L276)

13. Renumber figures in the order they are referenced in the text.

→ As suggested, the serial number of figures has been renumbered.

14. L299. What do authors mean by 'contamination' in this context?

→ To avoid ambiguity, the sentence has been changed and 'contamination' has been deleted. (L314-316)

15. L312, 337. Change to 'Gram-negative'.

→ It has been changed. (L344, 359-360)

16. L310 - 311. This title doesn't make sense. Maybe some words missing?

→ The title has been changed. (L357-358)

17. L364 - 365. What do authors mean by 'partial genotypes'?

→ The original intention was to express the synergistic inhibitory effect of PB in combination with CAZ against the biofilm formed by certain genotypes of strains. To avoid misunderstandings, 'partial genotypes' has been changed to 'certain genotypes', and the sentence has been modified. (L410-412, 417-420)

January 21, 2022

Dr. Xinlong He
Yangzhou University
Department of Pathogen Biology, School of Medicine
Yangzhou, Jiangsu 225009
China

Re: Spectrum01768-21R1 (Synergistic inhibitory effect of polymyxin B in combination with caftazidime against robust biofilm formed by *Acinetobacter baumannii* with genetic deficiency in Abal/AbaR quorum sensing)

Dear Dr. Xinlong He:

Thank you for submitting your manuscript to Microbiology Spectrum. As you will see your paper is very close to acceptance. Please modify the manuscript along the lines the reviewer has recommended. As these revisions are quite minor, I expect that you should be able to turn in the revised paper in less than 30 days, if not sooner. If your manuscript was reviewed, you will find the reviewers' comments below.

When submitting the revised version of your paper, please provide (1) point-by-point responses to the issues I raised in your cover letter, and (2) a PDF file that indicates the changes from the original submission (by highlighting or underlining the changes) as file type "Marked Up Manuscript - For Review Only". Please use this link to submit your revised manuscript. Detailed instructions on submitting your revised paper are below.

Link Not Available

Sincerely,

William Lainhart

Reviewer comments:

Reviewer #2 (Comments for the Author):

Overall the manuscript is much improved. However the figure legends are still lacking sufficient detail. These should enable interpretation of the figures separately from having to read the methods and results. Specific comments are below

-Figure 1 is still confusing. More detail needs to be provided in the legend to clarify. It is unclear what the differences are between B and D. What do the arrows indicate?

Figure 3 and 4. How is the data presented? mean +/- SD? I don't see any error bars in Figure 3?

-Figure 5 legend. Indicate what the differences indicated by * are compared to. Assuming it is to the untreated control, but this is not indicated.

-L227-229. This is still not clear. Is the definition of a group that it contains more than three strains?

Once these changes have been made, I do not need to re-review this manuscript.

Preparing Revision Guidelines

To submit your modified manuscript, log onto the eJP submission site at <https://spectrum.msubmit.net/cgi-bin/main.plex>. Go to Author Tasks and click the appropriate manuscript title to begin the revision process. The information that you entered when you first submitted the paper will be displayed. Please update the information as necessary. Here are a few examples of required

updates that authors must address:

- point-by-point responses to the issues I raised in your cover letter
- Upload a compare copy of the manuscript (without figures) as a "Marked-Up Manuscript" file.
- Each figure must be uploaded as a separate file, and any multipanel figures must be assembled into one file.
- Manuscript: A .DOC version of the revised manuscript
- Figures: Editable, high-resolution, individual figure files are required at revision, TIFF or EPS files are preferred

Please return the manuscript within 60 days; if you cannot complete the modification within this time period, please contact me. If you do not wish to modify the manuscript and prefer to submit it to another journal, please notify me of your decision immediately so that the manuscript may be formally withdrawn from consideration by Microbiology Spectrum.

Responses to Reviewers

Re: Spectrum01768-21R1 (Synergistic inhibitory effect of polymyxin B in combination with caftazidime against robust biofilm formed by *Acinetobacter baumannii* with genetic deficiency in AbaI/AbaR quorum sensing)
Authors: Yinyue Li, *et al.*

Dear Reviewer,

Thanks for your valuable comments and giving us the chance to further modify our manuscript. We have finished the point-by-point responses for this manuscript, and you will find the changes in the Marked Up Manuscript up-loaded.

Responses to Reviewer 2

Overall the manuscript is much improved. However the figure legends are still lacking sufficient detail. These should enable interpretation of the figures separately from having to read the methods and results. Specific comments are below

-Figure 1 is still confusing. More detail needs to be provided in the legend to clarify.

It is unclear what the differences are between B and D. What do the arrows indicate?

→ I think the reviewer is referring to Figure 2 rather than Figure 1 because the question described is related to Figure 2. B refers to matrix graph of correlation among gene expression, biofilm formation, and motility in representative MEM-resistant strains. D refers to matrix graph of correlation among gene expression, biofilm formation, and motility in representative MEM-resistant strains and MEM-sensitive strains. Arrows indicate that representative strains used for correlation analysis were derived from strains used for gene expression heat map analysis. As suggested, more detailed information have been provided for Figure 2 in the legend.

(Line 672-678)

Figure 3 and 4. How is the data presented? mean +/- SD? I don't see any error bars in Figure 3?

→ Figure 3 has been revised (Line 681). The data were presented as mean \pm SD of three biological replicates. Relevant changes have been made for Figure 3 and 4 in the legend. (Line 687-688, 694-696)

-Figure 5 legend. Indicate what the differences indicated by * are compared to.

Assuming it is to the untreated control, but this is not indicated.

→ As suggested, more detailed information have been provided in legend. Box with orange border refers to control without antibiotic treatment. All data were presented as means of three biological replicates. The significances were determined by nonparametric one-way ANOVA. * indicates significant increase at a P value of 0.05 when compared to control. * indicates significant decrease at a P value of 0.05 when compared to control. (Line 703-707)

-L227-229. This is still not clear. Is the definition of a group that it contains more than three strains?

→ The definition of a group is that it contains more than three strains. A clear description has been made at Line 227.

January 28, 2022

Dr. Xinlong He
Yangzhou University
Department of Pathogen Biology, School of Medicine
Yangzhou, Jiangsu 225009
China

Re: Spectrum01768-21R2 (Synergistic inhibitory effect of polymyxin B in combination with ceftazidime against robust biofilm formed by *Acinetobacter baumannii* with genetic deficiency in Abal/AbaR quorum sensing)

Dear Dr. Xinlong He:

Your manuscript has been accepted, and I am forwarding it to the ASM Journals Department for publication. You will be notified when your proofs are ready to be viewed.

EDITOR COMMENTS:

In final review of this manuscript, please ensure proper spelling of "ceftazidime" - there are multiple instances throughout the manuscript where it is misspelled as "caftazidime" including in the title.

Sincerely,

William Lainhart
Editor, Microbiology Spectrum
